# Impact of Influenza Vaccination on the Burden of Severe Influenza in the Elderly: Spain, 2017–2020

**DOI:** 10.3390/vaccines11061110

**Published:** 2023-06-17

**Authors:** Clara Mazagatos, Concepción Delgado-Sanz, Ana Milagro, María Liébana-Rodríguez, Amparo Larrauri

**Affiliations:** 1National Centre of Epidemiology, Institute of Health Carlos III, 28029 Madrid, Spain; cmazagatos@isciii.es (C.M.); cdelgados@isciii.es (C.D.-S.); 2CIBER Epidemiología y Salud Pública (CIBERESP), 28029 Madrid, Spain; 3Miguel Servet University Hospital, Microbiology, 50009 Zaragoza, Spain; amilagro@salud.aragon.es; 4Health Research Institute Aragón, 50009 Zaragoza, Spain; 5Servicio Medicina Preventiva, Hospital Universitario Virgen de las Nieves, 18014 Granada, Spain; maria.liebana.sspa@juntadeandalucia.es

**Keywords:** influenza, surveillance, burden, vaccination, evaluation

## Abstract

Annual influenza vaccination is the main strategy to reduce the burden of seasonal influenza epidemics and is recommended for the elderly in most countries with influenza vaccination strategies, with the main objective of preventing hospitalizations and mortality associated with seasonal influenza in this age group. Studies from different countries have estimated the benefits of seasonal influenza vaccination programs in the elderly, preventing a considerable number of cases, hospitalizations and deaths every year. A study measured the number of medically attended confirmed influenza cases in primary care that are prevented annually by vaccination in the population aged 65 and older in Spain, the Netherlands and Portugal, but estimates of the impact of the national influenza vaccination program in the prevention of severe disease in Spain are lacking. The two objectives of this study were to estimate the burden of severe influenza disease in the Spanish population and to measure the impact of influenza vaccination in the prevention of these outcomes in the population aged 65 years and older. Using influenza surveillance systems put in place before the COVID-19 pandemic, we conducted a retrospective observational study to estimate the burden of hospitalizations and ICU admissions in Spain between 2017–18 and 2019–20, by season and age group. Burden estimates for the 65+ group, combined with vaccine effectiveness (VE) and vaccination coverage (VC) data, were used as input data in an ecological, observational study to estimate the impact of the influenza vaccination program on the elderly. We found a higher burden of severe influenza disease in seasons 2017–18 and 2018–19, with A(H3N2) circulation, and in the youngest and oldest age groups. In those aged 65 and older, we estimated an average of 9900 influenza hospitalizations and 1541 ICU admissions averted by vaccination each year. Seasonal influenza vaccination was able to prevent between 11 and 26% influenza hospitalizations and around 40% ICU admissions in the elderly in the three pre-pandemic seasons. In conclusion, our study complements previous analyses in the primary care setting in Spain and demonstrates the benefits of the annual influenza vaccination program in the prevention of severe influenza disease in the elderly, even in seasons with moderate VE.

## 1. Introduction

Influenza viruses cause a significant annual burden of severe disease in younger and older adults. It is estimated that influenza infections are associated with over 5 million hospitalizations per year worldwide, [1] and with an annual excess of respiratory deaths between 4.0 and 8.8 per 100,000 individuals [2].

Seasonal influenza vaccination in Spain is indicated for healthcare professionals and specific target groups considered to be at risk of severe influenza disease, including populations aged 65 and older, pregnant women and the young population with underlying chronic conditions [3]. The aim of the influenza vaccination program is to reduce severe influenza outcomes and mortality in the populations most at risk, and influenza vaccines are offered to these groups free of charge. Influenza vaccination coverage in the Spanish population aged 65 and older remained constant in the last 10 years, ranging from 54 to 58% and experienced a considerable increase after the COVID-19 pandemic, reaching up to 67.7% in season 2020–21 [4]. However, as seen in other European countries [5], these figures are still far from the 75% goal for vaccine uptake in key risk groups set by the World Health Organization (WHO) [6]. To further increase vaccination uptake, an annual evaluation of the impact of seasonal influenza vaccination campaigns is key.

Two aspects might be considered regarding the low influenza vaccine coverage among the elderly, the main recommended group for vaccination. On the one hand, the widespread belief that influenza is a mild disease. On the other hand, the influenza vaccine effectiveness (VE) estimates, which are annually reported as low or moderate. Knowing the burden of severe disease caused by influenza in the elderly, as well as the actual number of severe influenza episodes averted by vaccination, is essential for disease risk communication and for understanding the true impact of influenza vaccination programs [7,8,9].

Different methodologies are available to estimate the impact of vaccination programs preventing influenza-related events in Europe and the US [10,11], but all have similarities and require the same input data: the number of events observed in the population, the effectiveness of the vaccine preventing those events, and the proportion of the population that is vaccinated.

The number of observed events in the population is the burden of disease, which can be estimated using influenza surveillance data. Before the COVID-19 pandemic, influenza surveillance in Spain consisted of a primary care network of sentinel physicians reporting cases of influenza-like illness, with a systematic selection of samples for testing and laboratory confirmation. For severe influenza surveillance, a network of hospitals reported confirmed influenza hospitalizations with certain severity criteria [12] to the surveillance system for Severe Hospitalized Confirmed Influenza Cases (SHCIC) that was implemented in Spain after the A(H1N1) influenza pandemic in 2009. Using surveillance data up to 2016, the highest rates of severe influenza hospitalizations in Spain were estimated in children under 5 years of age, and the population aged 65 years and older [13]. Those aged 65 and older also had the largest annual rate of deaths among influenza hospitalized patients (3 per 100,000 pop.) [13]. Since the 2017–18 influenza season, the system also began collecting the number of all confirmed influenza hospitalizations of any severity, providing a more comprehensive picture of the true burden of influenza hospitalizations in Spain. Influenza VE was estimated annually within the European I-MOVE network, with multicentre primary care and hospital-based test-negative case control studies [14,15,16,17]. National influenza vaccination coverage data in target groups are calculated and published annually by the Spanish Ministry of Health.

Effectively communicating the benefits of influenza vaccination to healthcare providers and the general population in terms of prevented burden of disease is essential to improve acceptability and increase uptake of seasonal influenza vaccines. An estimation of the averted number of medically attended confirmed influenza cases in the primary care setting in three European countries, including Spain, was published for the seasons from 2015–16 to 2017–18 [18], but impact estimates are not yet available for severe influenza outcomes in Spain.

In this study we aimed, first, to estimate the burden of severe influenza disease in the Spanish population, with information on influenza hospitalizations and ICU admissions reported to the Spanish influenza hospitalization surveillance system, during the 2017–18 to 2019–20 influenza seasons. Second, we aimed to measure the impact of influenza vaccination in the prevention of these outcomes, focusing on the population aged 65 and older, which is the main target group for influenza vaccination in Spain.

## 2. Materials and Methods

### 2.1. Study Design, Study Settings and Study Population

A retrospective observational study was conducted to estimate the burden of hospitalizations and ICU admissions in the Spanish population. We used data from the Spanish influenza hospitalization surveillance system, based on a network of public hospitals from all Spanish regions, covering 51–54% of the total Spanish population between the seasons of 2017–18 and 2019–20. Participating hospitals reported weekly aggregated data on all laboratory-confirmed influenza hospitalizations, and case-based data on those meeting severity criteria: pneumonia, acute respiratory distress syndrome, multiple organ failure, septic shock, ICU admission or death during admission. The population under surveillance was representative of the Spanish population and followed the same age-distribution: 4% <5 years, 10% 5–14 years, 65% 15–64 years, and 20% >64 years. To conduct the analysis of the impact of the vaccination program we used an ecological study design and restricted the study population to adults aged 65 and older.

### 2.2. Influenza Hospitalization and ICU rates

Cumulative rates of influenza-confirmed hospitalizations and ICU admissions were calculated using data from cases reported to the Spanish influenza hospitalization surveillance system in seasons of 2017–18, 2018–19 and 2019–20. To estimate rates by season and age group (<5, 5–14, 15–64 and >64 years), we used as denominators the reference catchment population, which was the sum of the population assigned to each of the participating hospitals, obtained from administrative records in each Spanish region. Taking into account the large variability in the estimated hospitalization rates by region, to combine data we followed an approach previously used [19] and applied a beta binomial model to control for the regional overdispersion, thus providing a better fit of the observed data, in order to calculate the national hospitalization rates. For this, the presence of overdispersion was previously assessed with the likelihood ratio test. The total numbers of hospitalized influenza cases were obtained by extrapolating cumulative rates to the Spanish population, by season and age group.

Influenza surveillance in hospitals was based on the reporting of all laboratory-confirmed positive cases, but no information was collected on influenza testing which can vary by age, time of the epidemic and hospital. Since the number of confirmed influenza hospitalizations reported depends directly on the swabbing policy in each participating hospital, this can lead to an underestimation of the real number of influenza hospitalizations. To avoid this, when estimating the burden of hospitalizations and ICU cases in the 65 and older group we further adjusted by using a correction factor, that is the proportion of patients aged 65 years and older who are swabbed for influenza diagnosis in Spanish hospitals. This proportion of swabbing was calculated from 2 hospitals participating in the I-MOVE+ (Integrated Monitoring of Vaccines in Europe) project, by linking data of clinical diagnoses indicative of acute respiratory infection from admissions and laboratory registries. The proportion of testing in patients aged 65 and older was estimated to be 40.5–42.7% in the three seasons included in this study (data not published). The estimated number of hospitalizations and ICU admissions in those 65 and older was used as input for the evaluation of the impact of the vaccination program in the elderly.

### 2.3. Impact of the Influenza Vaccination Program in the Elderly

We applied previously published methodology [10,18,20] to estimate the impact of the seasonal influenza vaccination program. Briefly, we compare the number of observed influenza hospitalizations and ICU admissions in the elderly population (n) to the estimated number that would occur without the vaccination program (N). To estimate N, we use the hospitalization burden obtained from surveillance data (n), vaccination coverage (VC) and influenza vaccine effectiveness (IVE) in the 65 and older population. For each season, we computed the number of averted events (NAE) as follows:$$NAE=N-n=\frac{n}{1-\left(VC\times VE\right)}-n=n\times \frac{VC\times VE}{1-(VC\times VE)}$$

We also computed by season the prevented fraction (PF), which is the proportion of events averted out of the total estimated events that would have occurred without influenza vaccination:PF = NAE/(n + NAE) × 100

As described earlier [18] to estimate the 95% CI for NAE and PF, we used a probabilistic Monte Carlo approach. We constructed empirical distributions for influenza-associated outcomes, positivity rate, IVE and VC and used the 2.5 and 97.5 percentiles of these empirical distributions to compute the 95% CI for NAE and PF. All analyses were performed using STATA software.

### 2.4. Input Data for the Estimation of Vaccination Impact

The number of observed events (n) was the estimated total number of influenza hospitalizations and ICU admissions in the population aged 65 years and older for each season. Influenza VC figures were obtained from the Ministry of Health, based on administrative data on the number of doses of influenza vaccine administered in the 65+ population, and are available online [4]. The IVE against influenza hospitalization was obtained from pooled European estimates from the I-MOVE multicentre hospital-based test-negative case control studies [15,16,17]. These studies were conducted in older hospitalized adults (≥65 years) and used a one-stage analysis of pooled individual data from SARI hospitalizations in different European study sites, with a varying number of hospitals participating each season. They provided seasonal VE estimates against influenza hospitalization by circulating subtypes. We used European IVE by type/subtype, pooling available estimates in seasons 2017–18 to 2019–20, and we weighted these estimates by the specific influenza type/subtype distribution each season in Spain. The IVE estimates used for impact estimations are shown in Table 1.

The IVE estimate against ICU admission in patients hospitalized with influenza was obtained from the literature. In a multicentre case–control study in Navarra, Spain, Casado et al. estimated a 74% (95% CI 42–88%) VE in preventing admissions to ICU in influenza hospitalized cases [21]. We used this estimate for the three seasons analysed in our study.

## 3. Results

### 3.1. Burden of Severe Influenza in Spain

#### 3.1.1. Cases Reported to the Spanish Influenza Hospitalization Surveillance System

The weekly number of influenza-confirmed hospitalizations and ICU admissions reported to the Spanish influenza hospitalization surveillance system and the dominant influenza viruses in the three seasons included in this study are shown in Figure 1. A total of 15,580 influenza-confirmed hospitalizations were reported in season 2017–18, 13,049 in season 2018–19, and 10,935 in season 2019–20. Among those hospitalized with confirmed influenza, 5977, 4919 and 3105 ICU admissions were reported in seasons 2017–18, 2018–19, and 2019–20, respectively. Circulation of B/A(H3N2), A(H1N1)pdm09/A(H3N2) and A(H1N1)pdm09/B influenza virus was dominant, respectively, in these influenza seasons.

#### 3.1.2. Estimated Burden of Influenza Hospitalizations and ICU Admissions in the Spanish Population

We estimated between 60 and 107 influenza hospitalizations per 100,000 population and between 4 and 6 ICU admissions per 100,000 population, with differences by season and age group (Table 2). Cumulative rates for both influenza hospitalization and ICU admission showed a U-shaped distribution in all seasons, with the highest rates in the youngest and oldest age group: 0–4 and 65+ years old.

In children under 5 years, we estimated between 165 and 238 influenza hospitalizations per 100,000 population and between 10 and 16 ICU admissions per 100,000 population, depending on the season. Likewise, in the elderly we estimated between 128 and 319 influenza hospitalizations and between 5 and 15 ICU admissions per 100,000 population (Table 2).

#### 3.1.3. Estimated Number of Influenza Hospitalizations and ICU Admissions, IVE Estimates and Influenza VC in the Elderly

Using the number of hospitalizations and ICU admissions in the Spanish population aged 65 and older (Table 2), and after adjusting for the proportion of patients swabbed, we estimated a burden of 69,563, 48,487 and 27,407 influenza hospitalizations and 3284, 2430 and 1061 ICU admissions among those aged 65 and older in seasons 2017–18, 2018–19 and 2019–20, respectively (Table 3). The number of observed events (n), VE estimates and influenza vaccination coverages used as inputs in the model for the estimation of vaccination impact in the elderly are shown in Table 3.

### 3.2. Impact of Influenza Vaccination in the Elderly

In the three seasons included in this study, influenza vaccination in the elderly prevented between 65 and 158 influenza hospitalizations and between 8 and 26 ICU admissions per 100,000 population in Spain (Table 4). On average, vaccination prevented 9900 hospitalizations and 1541 ICU admissions each season.

The prevented fraction range is 11–26% for influenza hospitalizations, and 40–41% for influenza ICU admissions (Table 4).

## 4. Discussion

Influenza was responsible for a considerable annual morbidity in Spain, with an annual range of 60–107 hospitalizations and 4–6 ICU admissions per 100,000 population, depending on the season. These results refer to seasons 2017–18, 2018–19 and 2019–20, a period when all seasonal influenza viruses (B, A(H3N2) and A(H1N1)pdm09) circulated in Spain. Our study describes the burden of severe influenza disease providing estimates of influenza hospitalizations and ICU admissions by age group during the three years prior to the COVID-19 pandemic in Spain. For the population aged 65 and older, which is the largest target group for seasonal influenza vaccination, we also present the impact of the vaccination program preventing these severe influenza outcomes, during the same period. Vaccination was able to prevent 11–26% hospitalizations and 40% ICU admissions with confirmed influenza infection in the elderly during the study period.

Previous analyses on the burden of influenza in Spain between 2010 and 2016 showed an annual average of 3616 severe influenza hospitalizations and 1232 ICU admissions for all ages [13]. Our estimates for seasons 2017–18 to 2019–20 are similar in terms of annual ICU admissions, but considerably higher for the burden of influenza hospitalizations, as we include for the first time all confirmed influenza hospitalizations in Spain regardless of the clinical severity.

The burden of severe influenza disease, in terms of influenza hospitalizations and ICU admissions, was considerably higher in seasons 2017–18 and 2018–19, with A(H3N2) circulation, than in season 2019–20, when influenza A(H1N1)pdm09 and B were dominant. Looking at global surveillance data from previous years, a high burden of disease is expected in the elderly in seasons with A(H3N2) circulation [22], and this is reflected in our study setting, where those aged 65 and older had the highest hospitalization rates in seasons 2017–18 and 2018–19, as opposed to 2019–20, when the most affected age group was children under 5 years. The U-shaped distribution of influenza hospitalizations by age has been described in previous estimates of severe hospitalizations in Spain [13] and in influenza burden estimates from other countries [7,23,24]. ICU rates in seasons 2017–18 and 2018–19 doubled the rates of season 2019–20 and were consistently higher in children under 5 years than in older age groups. All of these results are consistent with previous findings on seasonal influenza severity by type–subtype in Spain [25]. Notably, seasons with A(H3N2) circulation and high burden of disease in the elderly are usually associated with low VE estimates [26,27,28], which may be related, among other factors, to the high antigenic variability of A(H3N2) viruses that can lead to a mismatch between circulating and vaccine viral strains.

This study includes a three-season period prior to the COVID-19 pandemic, after which patterns of seasonal influenza circulation and disease severity may have changed. The emergence of the COVID-19 pandemic had a major impact on influenza surveillance, which was interrupted or adapted in many countries during the pandemic [29]. Additionally, there were important changes in healthcare seeking behaviors and in the organization of health systems. Many sentinel physicians were reallocated for the pandemic response, sentinel circuits for respiratory samples were modified and patient samples were sent to COVID-19 diagnostic centers [30]. In addition, a sharp decline in the circulation of influenza viruses and other respiratory pathogens was observed in Europe and globally during the winter of 2020–21 [31,32,33] with a progressive recovery in Spain after spring 2021 [34]. Since then, the changes observed in seasonal influenza patterns following COVID-19, together with changes in surveillance systems, call for a reassessment of the impact of influenza epidemics in the post-pandemic period, including estimates of the burden of disease from influenza and COVID-19 in the population. It is also possible that new influenza circulation patterns could lead to changes in influenza vaccination schedules, due to a variation in the usual temporal presentation of seasonal epidemics.

For the evaluation of a vaccination program, post-marketing monitoring and evaluation of vaccines through observational studies is essential to capture the global benefits of vaccination. In Spain, the transition towards integrated sentinel systems for the surveillance of acute respiratory infections after the pandemic will be key for this objective. When fully implemented, these systems will be able to provide annual estimates of the number of cases, hospitalizations and deaths caused by influenza and SARS-CoV-2 epidemics. In addition, by collecting vaccination data these surveillance systems provide estimates to monitor VE and to measure the impact of the influenza and COVID-19 vaccination strategies in Spain.

The impact of a vaccination program includes not only the direct effect of vaccines among those vaccinated, measured by vaccine efficacy results obtained in clinical trials, but also the indirect effects observed in the unvaccinated population, who also see their risk reduced without having received the vaccine. These indirect effects in the population are essential to understand why targeted influenza vaccination programs are such an effective public health intervention. Unlike other vaccination programs based on vaccines with higher efficacy, aiming to interrupt community transmission, the main goal of influenza vaccination programs is to reduce the burden of seasonal influenza disease on populations and health systems, especially severe influenza outcomes and influenza-related mortality in the populations most at risk. Therefore, to evaluate if a population-wide intervention such as influenza vaccination is being really effective, it is essential to quantify the vaccination impact in terms of averted severe influenza events, as explored in this paper in the Spanish context.

Despite the moderate or low efficacy of seasonal influenza vaccines when compared to other vaccines on the market, influenza vaccination programs have shown great benefits in terms of averted burden of disease in the population [8,18,19,35]. Our results on the burden of severe influenza prevented by vaccination complement previous results of averted mild influenza disease in the primary care setting in Spain [18] and add evidence to the beneficial impact of vaccination on the prevention of influenza hospitalizations in Portugal [8] and the US [36]. Similar to the primary care-based study in Spain [18], we found large differences in the averted burden of hospitalizations depending on the season and the dominant influenza viruses. The season with the largest burden and the largest NAE was 2017–18, when nearly 14,000 hospitalizations were prevented. However, in terms of PF, season 2019–20 showed the greatest impact with a 26% PF, meaning that influenza vaccination in Spain prevented more than 1 in 4 potential hospitalizations in the elderly. The impact of vaccination on ICU admissions was similar between seasons, which is probably influenced by the VE estimate used as an input in the model. As VE against ICU admission, we used a fixed value from the literature and applied it to the whole period, resulting in very similar PF estimates, around 40% in all three seasons. All these results suggest that vaccine effectiveness was the main driver for impact estimates and highlights the importance of having available VE estimates measuring protection against different outcomes. Input data on influenza burden, VC and VE should be carefully selected to match the time-period, population group and type of influenza event that is being evaluated as much as possible in order to ensure precision of the impact estimates and usefulness for public health.

Our study is subject to some limitations. Surveillance data did not provide information on the proportion of hospitalizations that were tested for influenza in Spanish hospitals. To control for potential underreporting, a correction factor was applied to estimate the burden in the elderly, but it could only be calculated with data from the two hospitals participating in the I-MOVE+ network, which may not be representative of the testing practices in the rest of the country. Observational studies measuring the impact of vaccination programs also have limitations that must be considered to properly interpret the results. First, the methodology used attributes all the observed benefit in a given period to the seasonal influenza vaccination program and does not allow a measure of the contribution of protection acquired in the population through either natural infection or previous vaccinations, which may be particularly relevant for a disease such as influenza, with high-incidence and a repeated annual vaccination schedule. Additionally, as mentioned before, indirect protection in the unvaccinated population is key to understanding the global impact of vaccination programs, but this methodology only estimates direct effects in those vaccinated assuming no herd immunity. Indirect protective effects in the elderly have been described, such as the prevention of influenza-associated mortality in contexts of extensive influenza vaccination in children due to a reduction in community transmissions of influenza [37,38]. These indirect effects are likely low in our study setting, as vaccination was not widely administered in children in Spain but should be addressed in future seasons. After the inclusion for the first time of healthy children aged 6–59 months in the 2022–23 influenza vaccination recommendations in Spain [39], indirect protection of vaccinating children in the reduction of severe influenza disease in the elderly may not be negligible. In this context, it is possible that impact studies based on VE and VC in the elderly might underestimate the global benefits of the influenza vaccination program in Spain.

## 5. Conclusions

This study demonstrates the benefits of the influenza vaccination program in Spain in the prevention of severe influenza episodes in the elderly population. Our estimates of the number of hospitalizations and ICU admissions averted by vaccination complement previous evidence on the prevention of mild influenza cases in Spain, providing a more comprehensive view of the annual burden of influenza-related morbidity prevented by influenza vaccination in the population aged 65 and older.

## Figures and Tables

**Figure 1 vaccines-11-01110-f001:**
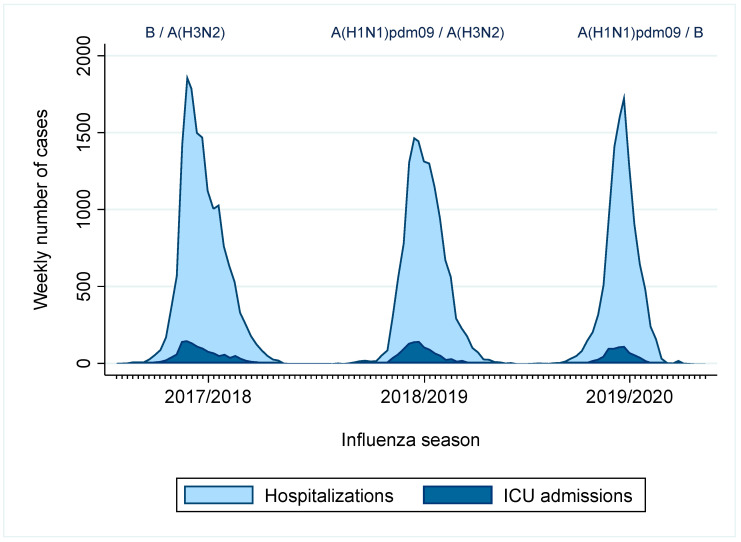
Weekly influenza hospitalizations and ICU admissions reported to the Spanish Influenza Hospitalization Surveillance System, for seasons 2017–18 to 2019–20.

**Table 1 vaccines-11-01110-t001:** Influenza circulation by type–subtype in Spain, and weighted vaccine effectiveness against influenza hospitalization.

	Distribution of Influenza Viruses Circulating in Spain (%)			Pooled IVE Weighted by Influenza Subtype Distribution in Spain
Season	A(H1N1)pdm09	A(H3N2)	B	VE (95% CI)
2017–18	7.6	25.3	64.9	30 (22; 38)
2018–19	36.3	61.2		20 (6; −36)
2019–20	59.5	19.0	21.5	49 (24; 66)

**Table 2 vaccines-11-01110-t002:** Cumulative rates and number of influenza hospitalizations and ICU admissions by age group and season, Spanish Influenza Surveillance System, seasons 2017–18 to 2019–20.

Influenza Hospitalizations							
Season	Age Group	Number	95% CI		Rate ^1^	95% CI	
2017–18	All ages	49,752	34,163	72,455	107.4	73.8	156.5
	<5 years	4893	2989	8009	237.2	144.9	388.3
	5–14 years	2398	1350	4258	49.6	27.9	88.1
	15–64 years	13,594	9117	20,270	44.5	29.8	66.3
	>64 years	28,173	20,463	38,789	318.7	231.5	438.8
2018–19	All ages	35,344	26,704	46,780	76.4	57.8	101.2
	<5 years	3266	2261	4717	164.8	114.1	238.0
	5–14 years	1815	1102	2990	37.6	22.8	61.9
	15–64 years	9635	6579	14,111	31.7	21.6	46.4
	>64 years	20,510	15,947	26,378	228.0	177.3	293.2
2019–20	All ages	27,657	21,212	36,060	59.9	45.9	78.1
	<5 years	4558	3152	6590	237.6	164.3	343.5
	5–14 years	1870	1233	2836	39.1	25.8	59.4
	15–64 years	9890	7481	13,075	32.6	24.7	43.1
	>64 years	11,703	8787	15,585	127.9	96.0	170.3
**ICU admissions**							
**Season**	**Age group**	**Number**	**95% CI**		**Rate ^1^**	**95% CI**	
2017–18	All ages	2881	2348	3535	6.2	5.1	7.6
	<5 years	321	210	490	15.6	10.2	23.8
	5–14 years	121	62	237	2.5	1.3	4.9
	15–64 years	1203	989	1463	3.9	3.2	4.8
	>64 years	1330	1087	1626	15.0	12.3	18.4
2018–19	All ages	2532	2211	2899	5.5	4.8	6.3
	<5 years	238	136	416	12.0	6.9	21.0
	5–14 years	93	59	147	1.9	1.2	3.0
	15–64 years	1186	1021	1379	3.9	3.4	4.5
	>64 years	1028	856	1234	11.4	9.5	13.7
2019–20	All ages	1763	1406	2213	3.8	3.0	4.8
	<5 years	197	124	314	10.3	6.5	16.4
	5–14 years	103	54	195	2.2	1.1	4.1
	15–64 years	1021	783	1332	3.4	2.6	4.1
	>64 years	453	357	575	5.0	3.9	6.3

^1^ Rates measured as influenza hospitalizations or ICU admissions per 100,000 population.

**Table 3 vaccines-11-01110-t003:** Estimated burden of influenza hospitalizations and ICU admissions, influenza vaccine effectiveness and vaccine coverage on the population aged 65 and older, Spain, seasons 2017–18 to 2019–20.

Season	Number Reported to SISSS	% Swabbing	Estimated Burden	VE ^1^; 95% IC (%)	VC ^2^ (%)
**Influenza hospitalizations**					
2017–18	28,173	40.5	69,563	30; 22–38	55.7
2018–19	20,510	42.3	48,487	20; 6–36	54.4
2019–20	11,703	42.7	27,407	49; 24–66	53.5
**ICU admissions**					
2017–18	1330	40.5	3284	74; 42–88	55.7
2018–19	1028	42.3	2430	74; 42–88	54.4
2019–20	453	42.7	1061	74; 42–88	53.5

^1^ VE: vaccine effectiveness against hospitalization (IMOVE estimates weighted by subtype distribution in Spain) and ICU admission (Casado et al. 2018). ^2^ VC: vaccine coverage in the population aged 65 and older (Spanish Ministry of Health).

**Table 4 vaccines-11-01110-t004:** Impact of influenza vaccination in the population aged 65 and older.

Season	2017–18			2018–19			2019–20		
	n	95% CI		n	95% CI		n	95% CI	
**Influenza hospitalizations**									
**Number averted**	13,985	9444	18,399	5892	817	11,173	9823	4078	14,924
**Rate (per 100,000 pop)**	158.2	106.8	208.1	65.5	9.1	124.2	107.4	44.6	163.1
**Prevented fraction (%)**	16.7	12.0	20.9	10.8	1.7	18.8	26.4	13.0	35.3
**ICU admissions**									
**Number averted**	2299	1033	3196	1628	736	2241	695	324	949
**Rate (per 100,000 pop)**	26.0	11.7	36.2	18.1	8.2	24.9	7.6	3.5	10.4
**Prevented fraction (%)**	41.2	23.9	49.2	40.1	23.4	47.9	39.6	23.3	47.1

## Data Availability

Data access policy within the National Epidemiological Surveillance Network (RENAVE) is similar to that of other public health agencies. The RENAVE is managed by the National Centre of Epidemiology (Instituto de Salud Carlos III), with the mandate to collect, analyse and disseminate surveillance data on infectious diseases in Spain. There is no public access to the RENAVE database, but influenza surveillance data are available upon request to Amparo Larrauri, head of the surveillance unit for influenza and other respiratory viruses at the Instituto de Salud Carlos III.

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
