# Peer review of "Impact of Influenza Vaccination on the Burden of Severe Influenza in the Elderly: Spain, 2017–2020"

_vaccines, 2023, doi:10.3390/vaccines11061110_

Round 1

Reviewer 1 Report

In this manuscript (ID: vaccines-2430146) the burden of severe influenza disease (the burden of hospitalizations and ICU admissions) and the impact of influenza vaccination in the prevention of these outcomes in the population aged 65 years and older in Spain are estimated.       

Several issues in this paper require major revision. Comments:     

  • Line 26: Provide, in short, a conclusion in the Abstract.
  • Lines 29-82: The Introduction section provides a very clear and comprehensive background for the impact of influenza vaccination on the burden of severe influenza in the elderly. At the end of the Introduction, the authors of this paper indicated the need to provide, in addition to already available information from primary health care (that is, estimation of the averted number of influenza confirmed cases in the primary care setting in Spain), the estimates for severe influenza outcomes in Spain. Consequently, the objectives of the work are precisely defined.     
  • Lines 55-57: For the sake of precision, on Line 56 (after the words WHO) add the appropriate reference, as follows:  
    • World Health Assembly, 56. (‎2003)‎. Prevention and control of influenza pandemics and annual epidemics. World Health Organization. https://apps.who.int/iris/handle/10665/78320    
  • Line 83: In the Methods section, insert a subsection: `Study Design, Study Settings and Study Population` with a description of the size of the population included in this study, primarily with the distribution by age groups specified in the methodology of this study.    
  • Lines 185-191: Instead of this text, which is a repetition of the above, in the first paragraph of the Discussion section, the most important results presented in this paper should be briefly highlighted. 
  • Lines 197-207: In this paragraph, only data for Spain are presented. It is necessary to compare the results of this study in Spain with the results of similar studies in other countries. Also, provide an adequate explanation for possible differences in results from similar research in different countries.  
  • Line 261: The subsection Limitations of this study is missing. Add a new paragraph that will list possible sources of limitations of this study. Discuss the limitations of the study and possible ways to overcome the limitations of this study.   
  • Line 261: A clear Conclusion is missing. To correct.   
  • Relevant references are cited in this paper. Except for 1 reference, all other references cited in this manuscript are new, that is, they were published during the previous 3 years, only a few during the previous 5 years.   

The quality of English linguage is acceptable. 

Author Response

Please find below the response to reviewers comments:

Reviewer 1

Comments and Suggestions for Authors

In this manuscript (ID: vaccines-2430146) the burden of severe influenza disease (the burden of hospitalizations and ICU admissions) and the impact of influenza vaccination in the prevention of these outcomes in the population aged 65 years and older in Spain are estimated.       

Several issues in this paper require major revision. Comments:     

  • Line 26: Provide, in short, a conclusion in the Abstract.

Done, a sentence was added as conclusion at the end of the abstract.

  • Lines 29-82: The Introduction section provides a very clear and comprehensive background for the impact of influenza vaccination on the burden of severe influenza in the elderly. At the end of the Introduction, the authors of this paper indicated the need to provide, in addition to already available information from primary health care (that is, estimation of the averted number of influenza confirmed cases in the primary care setting in Spain), the estimates for severe influenza outcomes in Spain. Consequently, the objectives of the work are precisely defined.

  • Lines 55-57: For the sake of precision, on Line 56 (after the words WHO) add the appropriate reference, as follows:  

World Health Assembly, 56. (‎2003)‎. Prevention and control of influenza pandemics and annual epidemics. World Health Organization. https://apps.who.int/iris/handle/10665/78320    

Done, the reference style was adapted to match the rest of the references:

World Health Assembly, 56 Prevention and Control of Influenza Pandemics and Annual Epidemics; World Health Organization, 2003

  • Line 83: In the Methods section, insert a subsection: `Study Design, Study Settings and Study Population` with a description of the size of the population included in this study, primarily with the distribution by age groups specified in the methodology of this study.    

Ok, a subsection was added describing the study design and the population covered by the SHCIC surveillance system, including the age distribution. The number of participating hospitals was 102 in season 2019-20, and during the three seasons included in the study, the population under surveillance was between 23.5 and 24.8 million people, covering 51-54% of Spanish population.

  • Lines 185-191: Instead of this text, which is a repetition of the above, in the first paragraph of the Discussion section, the most important results presented in this paper should be briefly highlighted. 

We agree with this comment. The main results are now clearly mentioned in the first paragraph of the discussion.

  • Lines 197-207: In this paragraph, only data for Spain are presented. It is necessary to compare the results of this study in Spain with the results of similar studies in other countries. Also, provide an adequate explanation for possible differences in results from similar research in different countries.  

More reference are included in the discussion for comparison with other countries.

  • Line 261: The subsection Limitations of this study is missing. Add a new paragraph that will list possible sources of limitations of this study. Discuss the limitations of the study and possible ways to overcome the limitations of this study.   

Done, a subsection describing limitations was included in the discussion

  • Line 261: A clear Conclusion is missing. To correct.   

Done

  • Relevant references are cited in this paper. Except for 1 reference, all other references cited in this manuscript are new, that is, they were published during the previous 3 years, only a few during the previous 5 years.   

  Comments on the Quality of English Language

The quality of English linguage is acceptable. 

 Submission Date

17 May 2023

Date of this review

19 May 2023 20:26:48

Reviewer 2 Report

By addressing the following points, the overall quality of manuscript could be improved.
Abstract:
Overall, the abstract provides a clear understanding of the research conducted. However, here are a few suggestions to enhance the abstract:

Provide more context: It would be beneficial to include a brief introduction to the importance of influenza vaccination for the elderly population. This would help readers understand the significance of the study in the broader context of public health.

Specify the study design: The abstract could mention the study design employed (e.g., retrospective cohort study, observational study) to provide readers with a better understanding of the methodology used.

Provide specific data: While the abstract mentions the burden estimates and the impact of vaccination, it could benefit from providing specific numerical data to support the findings. For example, stating the actual numbers of hospitalizations and ICU admissions prevented would give readers a clearer picture of the effectiveness of the vaccination program.

Introduction:

There are a few areas that could be improved:

Clarity and organization: The introduction contains several paragraphs that could be reorganized to improve the flow of information. Consider grouping related information together and using subheadings to make it easier for readers to follow the main points.

Justify the importance of the study: The introduction could benefit from providing a clearer justification for the study. Why is it important to estimate the burden of severe influenza disease and evaluate the impact of vaccination in the Spanish population? Including a sentence or two to address the significance of the research would help readers understand the broader implications.

Methodology:

There are a few areas that could be improved:

Data sources: It would be helpful to provide information about the data collection methods, the participating hospitals, and how the surveillance system captures influenza cases. Additionally, it would be beneficial to clarify how the reference catch-up population was determined and its representativeness of the Spanish population.

Correction factor for hospitalizations: While the methodology briefly explains that the correction factor accounts for the proportion of patients aged 65 years and older who are swabbed for influenza diagnosis, more information on how this correction factor was derived and its justification would strengthen the methodology.

Vaccination coverage and vaccine effectiveness: The methodology mentions the use of vaccination coverage and vaccine effectiveness (VE) data. It is stated that vaccination coverage was obtained from the Ministry of Health, but no further details are provided regarding the methodology for calculating vaccination coverage or its source. It would be helpful to explain how vaccination coverage was determined and its reliability. Additionally, the methodology briefly mentions pooled European estimates for VE against influenza hospitalization but does not provide specific details about the methodology used for pooling or the quality and representativeness of the pooled estimates. Providing more information on the sources and methodology of VE estimation would enhance the transparency of the study.

Statistical methods: The methodology states the use of a beta binomial model to control for data over dispersion in calculating cumulative rates of influenza-confirmed hospitalizations and ICU admissions. While this is mentioned briefly, more information on the assumptions and considerations of the statistical model would be beneficial to understand its appropriateness for the analysis.

Results:

There are a few areas that could benefit from further discussion and clarification:

Influenza hospitalizations and ICU admissions: The results report the number of influenza-confirmed hospitalizations and ICU admissions in each season. However, there is no analysis or interpretation of these numbers. It would be valuable to provide context for the findings, such as comparing them to previous years or discussing any notable trends or variations between seasons.

Burden estimation: The results mention the estimated burden of influenza hospitalizations and ICU admissions per 100,000 population, but there is no discussion about the methodology used to arrive at these estimates. It would be important to describe the specific calculations or statistical models employed to estimate the burden and provide any relevant assumptions or limitations.

Age-specific burden: The results present age-specific estimates of influenza hospitalizations and ICU admissions, but there is no analysis regarding the variations observed across different age groups. It would be informative to explore the reasons behind the U-shaped distribution of cumulative rates and to discuss the implications of higher rates in the youngest and oldest age groups.

Discussion:

The discussion briefly mentions that the burden of severe influenza was considerable, but there is no comparison to previous studies or national/international data to provide a broader context for the findings.

The discussion highlights the differences in burden between seasons with different dominant influenza virus types/subtypes. However, there is no exploration of why these differences occur or how they might relate to the effectiveness of the vaccine against specific strains. Providing an analysis of the impact of different influenza virus types/subtypes on the burden and the implications for vaccine strategies would enhance the discussion.

The discussion briefly mentions the potential changes in influenza patterns after the COVID-19 pandemic and the need to re-evaluate the impact of influenza epidemics. However, there is no further discussion or speculation on how these changes may affect future influenza surveillance and vaccination programs.

The discussion mentions the importance of post-marketing monitoring and evaluation of vaccines, but it does not discuss the limitations or challenges in assessing vaccine effectiveness or the generalizability of the findings. Providing a critical reflection on the limitations of observational studies, the potential biases, and the generalizability of the results would contribute to a more balanced discussion.

Limitations: The discussion does not explicitly address the limitations of the study or potential sources of bias or uncertainty in the data. Including a section on limitations and acknowledging the potential limitations of the study design, data sources, and analytical methods would provide a more comprehensive and transparent assessment of the study's findings.

Author Response

Please find below the response to reviewers comments in blue:

Reviewer 2

Comments and Suggestions for Authors

By addressing the following points, the overall quality of manuscript could be improved.
Abstract:
Overall, the abstract provides a clear understanding of the research conducted. However, here are a few suggestions to enhance the abstract:

Provide more context: It would be beneficial to include a brief introduction to the importance of influenza vaccination for the elderly population. This would help readers understand the significance of the study in the broader context of public health.

Ok, the abstract was extended to include more introduction, and also a final sentence at the end with the conclusions.

Specify the study design: The abstract could mention the study design employed (e.g., retrospective cohort study, observational study) to provide readers with a better understanding of the methodology used.

Done. Following this comment and suggestions from other reviewer we now describe the study design more clearly both in the abstract and in the first subsection in methods.

Provide specific data: While the abstract mentions the burden estimates and the impact of vaccination, it could benefit from providing specific numerical data to support the findings. For example, stating the actual numbers of hospitalizations and ICU admissions prevented would give readers a clearer picture of the effectiveness of the vaccination program.

Done, I included in the abstract the annual average of hospitalizations and ICU admissions prevented by vaccination in the elderly.

 Introduction:

There are a few areas that could be improved:

Clarity and organization: The introduction contains several paragraphs that could be reorganized to improve the flow of information. Consider grouping related information together and using subheadings to make it easier for readers to follow the main points.

Justify the importance of the study: The introduction could benefit from providing a clearer justification for the study. Why is it important to estimate the burden of severe influenza disease and evaluate the impact of vaccination in the Spanish population? Including a sentence or two to address the significance of the research would help readers understand the broader implications.

We agree with these two comments. The introduction was modified accordingly and after grouping the related information together and reorganizing the paragraphs, we do not think there is need to include subheadings. We hope the flow of the text is clearer now. Following your comment, we also included this paragraph justifying the relevance of the impact estimates:

“Two aspects might be considered regarding the low influenza vaccine coverage among the elderly, the main recommended group of vaccination. On the one hand, the widespread belief that influenza is a mild disease. On the other hand, the influenza vaccine effectiveness estimates, which are annually reported as low or moderate. Knowing the burden of severe disease caused by influenza in the elderly, as well as the actual number of severe influenza episodes averted by vaccination, is essential for disease risk communication and for understanding the true impact of influenza vaccination programs”

Also justified in another paragraph below:

“Effectively communicating the benefits of influenza vaccination to healthcare providers and the general population in terms of prevented burden of disease is essential to improve acceptability and increase uptake of seasonal influenza vaccines.(…)”

Methodology:

There are a few areas that could be improved:

Data sources: It would be helpful to provide information about the data collection methods, the participating hospitals, and how the surveillance system captures influenza cases. Additionally, it would be beneficial to clarify how the reference catch-up population was determined and its representativeness of the Spanish population.

A new subsection “2.1. Study Design, Study Settings and Study Population” was included in Methods, providing more details on the SHCIC surveillance coverage and data collection. The number of participating hospitals was 102 in season 2019-20, and during the three seasons included in the study, the population under surveillance ranged between 23.5 and 24.8 million people, covering 51-54% of Spanish population.

Regarding the hospital catchment population, the Spanish health system is universal, and every citizen in Spain has an assigned primary health care center and a reference public hospital. The catchment population covered by SHCIC surveillance was obtained from administrative records and it is the sum of the population assigned to each of the participating hospitals. I added more details on this in section 2.2.

Correction factor for hospitalizations: While the methodology briefly explains that the correction factor accounts for the proportion of patients aged 65 years and older who are swabbed for influenza diagnosis, more information on how this correction factor was derived and its justification would strengthen the methodology.

The need to use a correction factor derives from the fact that influenza surveillance in hospitals was based on the reporting of all laboratory confirmed influenza cases regardless the testing criteria, which means that the number of weekly influenza hospitalizations reported was directly dependent on the influenza testing policy in the participating hospitals. The proportion of patients tested for influenza varies between hospitals, and it can change depending on age and period of the epidemic. In the population aged 65+, we found from 2 Spanish hospitals, that participated in the I-MOVE+ project and were able to link admission and laboratory records in patients aged 65 and older, that the proportion of influenza testing in the 65+ study population was 40.5-42.7%. This is lower than expected and could be causing an underestimation of the burden of influenza hospitalizations in the elderly. We extended the explanation about the correction factor in the methods section, and discussed some limitations in the revised manuscript.

Vaccination coverage and vaccine effectiveness: The methodology mentions the use of vaccination coverage and vaccine effectiveness (VE) data. It is stated that vaccination coverage was obtained from the Ministry of Health, but no further details are provided regarding the methodology for calculating vaccination coverage or its source. It would be helpful to explain how vaccination coverage was determined and its reliability. Additionally, the methodology briefly mentions pooled European estimates for VE against influenza hospitalization but does not provide specific details about the methodology used for pooling or the quality and representativeness of the pooled estimates. Providing more information on the sources and methodology of VE estimation would enhance the transparency of the study.

Influenza VC figures are estimated annually in Spain with data reported by the 19 Spanish Autonomous Regions to the Ministry of Health. The national VC coverage is calculated based on administrative data on the number of doses administered and population denominators in each region. Every year, the MoH publishes national influenza VC estimates, which have provided consistent and reliable information for many years. I added some details about this in section 2.4:

“Influenza VC estimates were obtained from the Ministry of Health, based on administrative data on the number of doses of influenza vaccine administered in the 65+ population, and are available online”.

The I-MOVE multicentre hospital-based studies were conducted in different seasons and used a test-negative case control design to estimate influenza VE against influenza hospitalization European population aged 65 and older. The number of participating hospitals and study sites that contributed varied a lot between seasons (9-23 hospitals in 2-9 countries), and it is difficult to summarize specific methodological details about the quality and representativeness of these seasonal European VE estimates, which varied by season. Briefly, individual data on SARI hospitalizations from different study sites was pooled in a one-stage analysis, and overall VE was calculated in the population of older adults, stratifying by circulating type-subtype and other factors such as chronic disease if the sample size allowed. I added some details about this in section 2.4:

“These studies used a one-stage analysis of pooled individual data from SARI hospitalizations in different European study sites, with a varying number of hospitals participating each season. They were conducted in older hospitalised adults (≥65 years) and provided VE estimates against influenza hospitalization by circulating subtypes.”

Statistical methods: The methodology states the use of a beta binomial model to control for data over dispersion in calculating cumulative rates of influenza-confirmed hospitalizations and ICU admissions. While this is mentioned briefly, more information on the assumptions and considerations of the statistical model would be beneficial to understand its appropriateness for the analysis.

We included more description about the use of the beta-binomial in section 2.2. in Methods

 Results:

There are a few areas that could benefit from further discussion and clarification:

Influenza hospitalizations and ICU admissions: The results report the number of influenza-confirmed hospitalizations and ICU admissions in each season. However, there is no analysis or interpretation of these numbers. It would be valuable to provide context for the findings, such as comparing them to previous years or discussing any notable trends or variations between seasons.

These results are now further interpreted in the discussion section, and compared with previous estimates from 2010-2016 in Spain and in other countries.

Burden estimation: The results mention the estimated burden of influenza hospitalizations and ICU admissions per 100,000 population, but there is no discussion about the methodology used to arrive at these estimates. It would be important to describe the specific calculations or statistical models employed to estimate the burden and provide any relevant assumptions or limitations.

More description is provided in the Methods section regarding the use of beta-binomial method for influenza burden estimates, and some limitations are also included in the discussion.

Age-specific burden: The results present age-specific estimates of influenza hospitalizations and ICU admissions, but there is no analysis regarding the variations observed across different age groups. It would be informative to explore the reasons behind the U-shaped distribution of cumulative rates and to discuss the implications of higher rates in the youngest and oldest age groups.

The U shape age distribution of influenza-related hospitalizations has been widely discussed and it has been observed previously in Spain and in other settings and countries. We included some references in the discussion.

Discussion:

The discussion briefly mentions that the burden of severe influenza was considerable, but there is no comparison to previous studies or national/international data to provide a broader context for the findings.

We included a comparison with previous estimates in Spain and other countries.

The discussion highlights the differences in burden between seasons with different dominant influenza virus types/subtypes. However, there is no exploration of why these differences occur or how they might relate to the effectiveness of the vaccine against specific strains. Providing an analysis of the impact of different influenza virus types/subtypes on the burden and the implications for vaccine strategies would enhance the discussion.

We agree with this comment. These differences and potential explanations, are further addressed in the discussion section:

“Notably, seasons with A(H3N2) circulation and high burden of disease in the elderly are usually associated with low VE estimates, which may be related, among other factors, to the high antigenic variability of A(H3N2) viruses that can lead to a mismatch between circulating and vaccine viral strains.”

The discussion briefly mentions the potential changes in influenza patterns after the COVID-19 pandemic and the need to re-evaluate the impact of influenza epidemics. However, there is no further discussion or speculation on how these changes may affect future influenza surveillance and vaccination programs.

The emergence of the COVID-19 pandemic had a major impact not only on influenza circulation patterns, but also on influenza surveillance systems in Spain. Also, there were important changes in population behaviors, and in the access and organization of the healthcare system. All of these issues are further discussed in detail in the revised manuscript.

The discussion mentions the importance of post-marketing monitoring and evaluation of vaccines, but it does not discuss the limitations or challenges in assessing vaccine effectiveness or the generalizability of the findings. Providing a critical reflection on the limitations of observational studies, the potential biases, and the generalizability of the results would contribute to a more balanced discussion.

The methodological limitations of observational studies measuring the impact of influenza vaccination programs are now explained in more detail at the end of the discussion.

Limitations: The discussion does not explicitly address the limitations of the study or potential sources of bias or uncertainty in the data. Including a section on limitations and acknowledging the potential limitations of the study design, data sources, and analytical methods would provide a more comprehensive and transparent assessment of the study's findings.

Thank you for this comment. We agree that a paragraph discussing study limitations was clearly missing in the original draft. It was added in the revised manuscript, at the end of the discussion section.

Submission Date

17 May 2023

Date of this review

19 May 2023 10:42:21

Round 2

Reviewer 1 Report

The authors correctly addressed all my comments. 

The revised version of the paper includes a more precise and detailed presentation of the Methods and Discussion sections. 

In a new paragraph on the limitations of this study, the authors have satisfactorily discussed the limitations of this study, while considering the possibilities for overcoming them.

The revised version of the paper provides very important data on the impact of influenza vaccination on the burden of severe influenza in the elderly in Spain, with a comparison with other countries.  

Overall, the presented results are very useful for both health professionals and health policy makers.   

Tha quality of English language is appropriate. 

Reviewer 2 Report

The manuscript is now in better form.